# Webly Supervised Fine-Grained Image Recognition with Graph Representation and Metric Learning

Jianman Lin [1], Jiantao Lin [2], Yuefang Gao [3,*], Zhijing Yang [4] and Tianshui Chen [4]

1    School of Mechanical and Electrical Engineering, Guangdong University of Technology,
     Guangzhou 510006, China
2    School of Intelligent Science and Engineering, Jinan University, Zhuhai 519077, China
3    College of Mathematics and Informatics, South China Agricultural University, Guangzhou 510642, China
4    School of Information Engineering, Guangdong University of Technology, Guangzhou 510006, China
*    Correspondence: gaoyuefang@scau.edu.cn

**Abstract:** The aim of webly supervised fine-grained image recognition (FGIR) is to distinguish sub-ordinate categories based on data retrieved from the Internet, which can significantly mitigate the dependence of deep learning on manually annotated labels. Most current fine-grained image recognition algorithms use a large-scale data-driven deep learning paradigm, which relies heavily on manually annotated labels. However, there is a large amount of weakly labeled free data on the Internet. To utilize fine-grained web data effectively, this paper proposes a Graph Representation and Metric Learning (GRML) framework to learn discriminative and effective holistic–local features by graph representation for web fine-grained images and to handle noisy labels simultaneously, thus effectively using webly supervised data for training. Specifically, we first design an attention-focused module to locate the most discriminative region with different spatial aspects and sizes. Next, a structured instance graph is constructed to correlate holistic and local features to model the holistic–local information interaction, while a graph prototype that contains both holistic and local information for each category is introduced to learn category-level graph representation to assist in processing the noisy labels. Finally, a graph matching module is further employed to explore the holistic–local information interaction through intra-graph node information propagation as well as to evaluate the similarity score between each instance graph and its corresponding category-level graph prototype through inter-graph node information propagation. Extensive experiments were conducted on three webly supervised FGIR benchmark datasets, Web-Bird, Web-Aircraft and Web-Car, with classification accuracy of 76.62%, 85.79% and 82.99%, respectively. In comparison with Peer-learning, the classification accuracies of the three datasets separately improved 2.47%, 4.72% and 1.59%.

**Keywords:** webly supervised learning; fine-grained image recognition; graph representation learning; graph metric learning; noisy data

## 1. Introduction

Aiming at distinguishing sub-ordinate categories, FGIR has attracted increasing attention because it benefits various applications ranging from daily life to intelligent industry. Most algorithms currently use a deep learning paradigm driven by high-quality data to distinguish subclasses, which relies heavily on large-scale, manually labeled data. Therefore, it may not be possible to apply them to realistic applications. Alternatively, there is a large amount of weakly labeled data on the Internet that can be used to train models to alleviate the reliance of current fine-grained recognition algorithms on human annotation. To this end, much work has been devoted to solving fine-grained image recognition in webly supervised scenarios [1,2]. However, the inclusion of a certain percentage of noisy labels in the web-retrieved data can adversely affect the training of the model. In addition, the inherent small inter-class variance and large intra-class variance in fine-grained images further increase the recognition difficulty [1].

To address the existing challenges, it is necessary to learn effective feature representations from noisy data that can characterize fine-grained images, following two important aspects: samples from different subclasses are always similar in the whole and the differences between them are usually located in specific local regions. Therefore, it is crucial to accurately locate discriminative local regions. On the other hand, noisy labels can seriously impair model training. Thus, it is critical to correct incorrectly annotated samples and exclude out-of-distribution (OOD) samples. To help discover noisy labels, current webly supervised learning works rely on label or feature consistency to discover noisy labels [2]. Recent algorithms also adapt these strategies to address the webly supervised FGIR task and achieve positvie progress [1]. However, all these works merely utilize holistic features while ignoring local information that is crucial for the FGIR scenarios, limiting their performance in real-world applications.

In this paper, a graph representation and metric learning framework is proposed to learn instance-level and category-level graph representations to capture the holistic–local information of each instance and category and their interactions, while evaluating their similarity to help correct for noisy labels and discover OOD samples. The proposed framework first exploits an attention module to automatically extract discriminative local regions from the image and uses the holistic features with these local regions to initialize the nodes of the instance graph. Second, introducing graph prototypes with the same structure as the instance graph for each category, which are updated by moving average using the instance graph according to the category of the samples. Finally, the graph matching module is used to explore the interaction between holistic and local features through the propagation of graph node information on the one hand and to measure the similarity between the instance graph and the corresponding category graph prototype to effectively help correct the noisy labels and remove OOD samples on the other hand.

## 2. Related Work

### 2.1. Fine-Grained Image Recognition

The task of fine-grained image recognition is to distinguish sub-ordinate categories. This is challenging, mainly because inter-class differences between fine-grained classes tend to be subtle and localized, while intra-class differences may appear large due to differences in gesture and color [1]. Therefore, a key to fine-grained recognition is to discover and represent discriminative local regions. In recent years, a large number of research works have proposed to use important local regions of images to improve the recognition ability of fine-grained images. Roughly speaking, these works can be divided into two groups. The first group is to use artificially additionally annotated bounding boxes and components to localize discriminative regions in fine-grained images, such as using artificial localization of local appearance features likes face and eyes and combine them with global features for breed classification of dogs [3]. Experimental results show that precise positioning can provide more effective fine-grained features, which can greatly improve classification performance. The second group uses an unsupervised way to automatically locate discriminative regions in fine-grained images. For example, a reinforcement learning-based fully convolutional attention local network is proposed to adaptively focus on discriminative regions in images and the greedy reward strategy for image-level labels is used to train the framework to obtain better recognition result [4]. These algorithms rely on deep neural networks that require large-scale manual annotation data. However, the cost of collecting such data is very high, especially for fine-grained images, which require specialized knowledge to accurately label. In contrast, there are many weakly labeled images on the Internet, which can be used to optimize fine-grained recognition models without manual annotation.

### 2.2. Webly Supervised Learning

The aim of webly supervised learning is to capture effective feature representation from free web data. However, it is a challenge due to label noise and data bias [1]. In

recent years, learning from networked data has become increasingly popular and many works have been dedicated to solving this problem. Ref. [5] introduces a deep denoising network that combines bag-level MIL and attention-based instance-level MIL to filter out noise in web-supervised datasets, but such method cannot be trained end-to-end, which limits the application in practical scenarios. Ref. [6] utilizes a combination of multiple instance learning (MIL) and memory modules to solve label noise and background noise in noisy data. Ref. [2] employs momentum prototypes and contrast loss for label correction and Ref. [6] proposes to learn category-level feature-consistent representations through image-level feature contrast loss to help correct nosiy labels. However, few of these works have been specifically designed for fine-grained image recognition of scientific importance and application. To the best of our knowledge, the only existing work [1] proposes to cross update two parallel networks using "easy" and "hard" examples, thus alleviating the cumulative error during training the fine-grained recognition models with webly learning. Different from these works, we propose a graph representation and metric learning (GRML) framework to capture category-level holistic–local features by learning corresponding graph prototypes for each class, while using the learned category-level graph prototype features to help correct noisy labels and outlier samples, so as to achieve end-to-end processing of noisy data and training of classification model.

## 3. Methodology

In this section, we introduce the graph representation and metric learning (GRML) framework, which models the holistic–local information for each instance by learning the graph representation and builds graph prototype for each category to model the holistic–local information, as well as helping to correct noise labels and exclude OOD samples by measuring similarities between graph representation and graph prototypes. As shown in Figure 1, For a given sample input $\{x_i, y_i\}$, where $x_i$ is the input image data and $y_i$ is the image label. First, we obtain the holistic features using the feature extractor (CNN), then we extract the discriminative local regions from the image using the attention module and the holistic features associated with these local regions are used to initialize the instance graph nodes. At the same time, a graph prototype is introduced for each category and updated by moving average. Finally, the graph matching module is used to mine the holistic–local feature interaction through the propagation of graph node information and measuring the similarity between the instance graph and the corresponding category graph prototype, then evaluate their similarities to effectively help correct the noisy labels and exclude OOD samples.

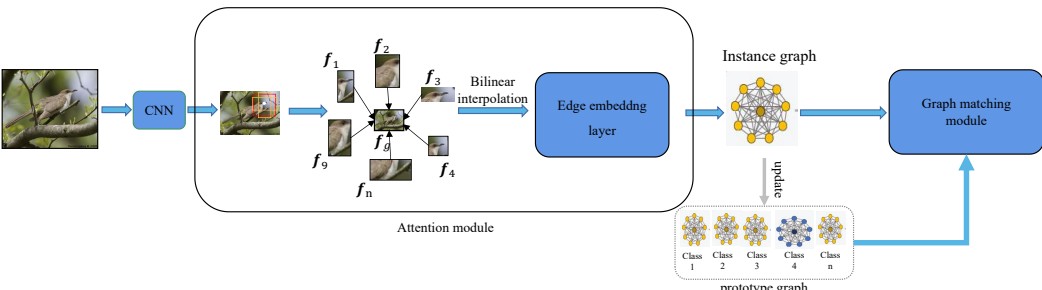

**Figure 1.** An overall illustration of the proposed GRML framework. It consists of two crucial modules, i.e., attention-focused and graph matching modules. The attention-focused module first locates local regions with different sizes and aspects that cover the most discriminative contents. Meanwhile, the graph matching module is used to explore information interactions to learn instance-level and category-level graph representation, as well as their matching to correct the noisy labels and exclude OOD samples.

### 3.1. Attention Module

As mentioned above, discriminative local regions play a key role in fine-grained image recognition. Inspired by [7], areas with larger response values in the feature map are more likely to be discriminant parts. We propose to utilize an attention module to automatically locate and extract the discriminative region features of images. To be more specific, we pass the holistic image features $f_g$ obtained from the feature extractor (CNN) through a convolutional layer with a convolutional kernel size of $3 * 3$ and all parameters of $1/9$ to compute a mean filtered feature map $f'_g$ and then calculate the mean value of each position $(w, h)$ based on the number of channels in the feature map $f'_g$ to obtain the feature map $f_g^{avg}$. Finally, we search for the maximum response value area in the feature map $f_g^{avg}$ and locate its coordinates as $(i, j)$; the specific formula is as follows:

$$f_g^{avg} = \frac{\sum_{n=1}^{C} f'_g(:,:,n)}{C} \tag{1}$$

$$(i, j) = \underset{w,h}{argmax} f_g^{avg} \tag{2}$$

Here, $W$, $H$ and $C$ are the width, height and channel number of the feature map $f'_g$ and $\underset{w,h}{argmax}$ represents the row and column corresponding to the search for the maximum value. Then, based on the obtained coordinate positions $(i, j)$, multiple local regions with varying area sizes and aspect ratios are intercepted in the feature map $f_g$ which is centered on it. In this experiment, three different area sizes $S_1, S_2, S_3$ and three different aspect ratios $A_1, A_2, A_3$ are set. Finally, a set of regions $R$ will be generated to fully capture information about discriminating subtle locations in the image.

$$R = \{r_1, r_2, r_3, r_4, r_5, r_6, r_7, r_8, r_9\} \tag{3}$$

To facilitate the construction of subsequent graphs and data processing, the $R$ local regions of different sizes are transformed into the same dimension $W * H * C$ by bilinear interpolation. Then, using adaptive global average pooling to reduce the dimension of holistic features and local features to the C-dimension feature vector as the node embeddings $V_{X_i} = \{f_1, f_2, f_3, f_4, f_5, f_6, f_7, f_8, f_9, f_g\}$. In addition, we use an edge embedding layer to explore the relationship between two adjacent nodes as the edge embeddings $e_{x_i,m,n}$, where $m$ and $n$ represent the indexes of two adjacent nodes. Finally, an instance graph $G_{x_i} = \{V_{x_i}, E_{x_i}\}$ is constructed, where $E_{x_i} = \{e_{x_i,m,n}\}_{m,n=0}^{M-1}$ denotes the edge embeddings of any two adjacent nodes in the graph. The specific formula is as follows:

$$e_{x_i,m,n} = f_{edge}(v_{x_i,m}||\hat{m}||v_{x_i,n}||\hat{n}) \tag{4}$$

Here, $\cdot||\cdot$ represents the connection operation of two vectors, $f_{edge}()$ is the edge embedding layer, $\hat{m}$ and $\hat{n}$, respectively, which represents the one-hot vector with subscript $m$ and $n$ being 1. The introduction of $\hat{m}$ and $\hat{n}$ aims to construct the spatial structure between global-local features.

### 3.2. Graph Prototype

To correct the noise labels and exclude OOD samples so as to alleviate the impact of noise data on network training, a category-level graph prototype is introduced for each class. For each category K, a graph prototype $G_k = \{V_k, E_k\}$ is constructed with the same structure as the instance graph and is initialized to 0. To better learn the feature consistency of each category, the graph prototype is updated by moving the average in the subsequent training process. The specific update method is as follows:

$$V_k \leftarrow mV_k + (1 - m)V_{x_i} \quad \forall i \in \{i|y_i = k\} \tag{5}$$

$$E_k \leftarrow mE_k + (1-m)E_{x_i} \quad \forall i \in \{i|y_i = k\} \tag{6}$$

Here, $m$ is the weight coefficient, which is set to 0.999 in the experiment.

### 3.3. Graph Matching Module

For input samples of the same category, their features have common characteristics. Inspired by [8], a graph matching module is introduced in this paper to propagate information on instance graph nodes and measure the graph-level similarity between instance graphs and their corresponding category graph prototypes. As shown in Figure 2, the graph matching module consists of a graph propagation layer $f_{\theta_p}()$, a graph interaction layer $f_{\theta_i}()$, a graph update layer $f_{\theta_u}()$ and a graph aggregation layer $f_{\theta_a}()$, where $\theta_p$, $\theta_i$, $\theta_u$ and $\theta_a$, respectively, represents the parameters of each layer of the network. Through the graph matching module, the intra-graph and inter-graph information of each instance sample and its corresponding class graph prototype can be fully utilized and the similarity between them can be measured, so as to learn a more category-discriminative graph-level feature representation.

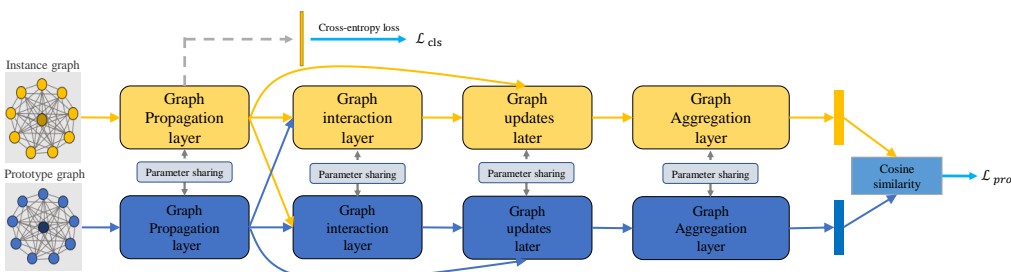

**Figure 2.** An overall illustration of the graph matching module. The graph matching module is used to propagate information to the nodes of the instance graph and measure the graph-level similarity of the instance graph to the corresponding category diagram prototype.

### 3.3.1. Graph Propagation Layer

To make full use of intra-graph information for each graph [9,10], we compute an intra-graph node representation for each node within the graph by propagating all its neighbor node embeddings along the edge embedding to explore information interaction within each graph. Specifically, take $G_{x_i} = \{V_{x_i}, E_{x_i}\}$ as an example; for each node $m \in [0, M-1]$, we first concatenate its node features $v_{x_i,m}$ with the adjacent node $v_{x_i,n}$ and their edge embedding $e_{x_i,m,n}$ together, then the nodes information propagates in the graph through a graph propagation layer $f_\theta()$. Finally, all neighborhood information are aggregated as $v_{x_i,m}^{intra}$ in the graph by an average manner; the specific formula is as follows:

$$v_{x_i,m}^{intra} = \frac{1}{M} \sum_{n \in [0,M-1]} f_{\theta_p}(v_{x_i,m}||v_{x_i,n}||e_{x_i,m,n}) \tag{7}$$

For each category of graph prototypes, the same method is used to obtain the node representation $v_{k,m}^{intra}$ after the propagation of node information in the graph. After the instance graph passes through the graph propagation layer to complete the interaction between the holistic and local features, one of the branches concatenates the features of nodes from the instance graph and uses a fully connected layer to obtain the instance-level feature representation and feed them into the classifier for classification prediction.

### 3.3.2. Graph Interaction Layer

To make full use of the cross-graph information, we introduce a graph interaction layer to measure how well a node in one graph (e.g., $G_{x_i}$) matches one or more nodes in another graph (e.g., $G_k$) and then calculate a cross-graph node representation $v_{x_i,m}^{cross}$ for each node $v_{x_i,m}^{intra}$. Specifically, we first consider graphs $G_{x_i}$ and $G_k$ as inputs to the graph interaction

layer. Then, the similarity score of nodes across graphs is calculated by computing the inner product of their node features $v_{x_i,m}^{intra}$ and $v_{k,n}^{intra}$, where $m, n \in [0, M - 1]$. Finally, the similarity score is regarded as a weight assignment, resulting in a new cross-graph node express $v_{x_i,m}^{cross}$ and $v_{k,m}^{cross}$. The specific formula is as follows:

$$v_{x_i,m}^{cross} = \sum_n v_{k,n}^{intra} \frac{\exp(v_{x_i,m}^{intra} \cdot v_{k,n}^{intra})}{\sum_n \exp(v_{x_i,m}^{intra} \cdot v_{k,n}^{intra})} \tag{8}$$

$$v_{k,n}^{cross} = \sum_m v_{x_i,m}^{intra} \frac{\exp(v_{x_i,m}^{intra} \cdot v_{k,n}^{intra})}{\sum_n \exp(v_{x_i,m}^{intra} \cdot v_{k,n}^{intra})} \tag{9}$$

### 3.3.3. Graph Update Layer

Following the graph propagation layer and graph interaction layer, the node feature $v_{x_i,m}^{intra}$ after information exchange in the graph and the node feature $v_{x_i,m}^{cross}$ through information propagation between graphs can be obtained. Then, a graph update layer $f_{\theta_u}()$ is used to fuse the information after intra-graph propagation and inter-graph propagation. Specifically, we concatenate information from the graph propagation and graph interaction layers and pass it through the graph update layer to obtain the graph node feature representation after the fusion of graphs $G_{xi}$ and $G_k$. The formula is as follows:

$$v_{x_i,m}^{update} = f_{\theta u}(v_{x_i,m}^{intra} || v_{x_i,m}^{cross}) \tag{10}$$

$$v_{k,n}^{update} = f_{\theta u}(v_{k,n}^{intra} || v_{k,n}^{cross}) \tag{11}$$

### 3.3.4. Graph Aggregation Layer

To measure the similarity at the graph level, the nodes of graph $G_{x_i}$ and graph $G_k$ are aggregated, respectively, after the graph update layer; specifically, we concatenate all the node information in the graph and pass it through the graph aggregation layer to obtain the aggregated information, and the aggregated feature vectors are encoded as feature vector representations $f_{x_i}$ and $f_k$; the specific formula is as follows:

$$f_{x_i} = f_{\theta a}(v_{x_i,0}^{update} || v_{x_i,0}^{update} \cdots v_{x_i,m}^{update}) \tag{12}$$

$$f_k = f_{\theta a}(v_{k,0}^{update} || v_{k,0}^{update} \cdots v_{k,m}^{update}) \tag{13}$$

### 3.3.5. Similarity Measure

Finally, the cosine similarity between the two feature vectors $f_{x_i}$ and $f_k$ is calculated with the following formula to measure the similarity score between the graph $G_{x_i}$ and the graph $G_k$:

$$S_k = cosine(f_{x_i}, f_k) = \frac{f_{x_i} \cdot f_k}{||f_{x_i}|| \cdot ||f_k||} \tag{14}$$

After repeating the above operation $C$ times, where $C$ is the number of categories, the similarity $S = \{s_1, s_2, s_3, ..., s_c\}$ between the input sample $x_i$ and all category graph prototypes can be obtained. If the category of the graph prototype $G_k$ is the same as the category of the instance graph $G_{x_i}$, the similarity score $S_k$ will be close to 1; otherwise, it will be close to 0. So it can be expressed as a graph matching task and a graph matching loss function is introduced for training as follows:

$$\mathcal{L}_{pro}^i = \sum_{k=1}^{K} l_{pro}^{ins,k} \tag{15}$$

$$l_{pro}^{ins,k} = \begin{cases} 1 - S_k, & y_i = k \\ 1 + S_k, & otherwise \end{cases} \tag{16}$$

### 3.4. Noise Correction

This paper adopts a simple and effective method to correct noisy labels and remove OOD samples during training. For each instance sample, a pseudo-label is obtained by combining the classifier output probability distribution $p_i$ with $d_i$, where $d_i$ is the class probability distribution after the similarity score $S$ is normalized by Softmax. The formula is as follows:

$$q_i = \alpha p_i + (1 - \alpha) d_i \tag{17}$$

$$d_i^k = \frac{\exp(s_k/\tau)}{\sum_{k=1}^{K} \exp(s_k/\tau)} \tag{18}$$

Here, $\tau$ is a temperature coefficient and the noise label correction and OOD samples are removed according to the following rules: (1). If the maximum score of $q_i$ is higher than the set threshold $T$, the category with the highest score is used as the pseudo-label; (2). If the score of the original label $C$ is higher than the average probability of the class, the original label is retained; (3). In the rest of the cases, it is marked as an outlier sample. It is expressed in detail as follows:

$$y_k' = \begin{cases} argmax_k \quad q_i^k, & \text{if } max_k \quad q_i^k > T, \\ y_i, & \text{else if } q_i^{y_i} > 1/k, \\ \text{OOD}, & \text{otherwise.} \end{cases} \tag{19}$$

### 3.5. Optimization

In addition to the graph matching loss function, this paper also uses a categorical cross-entropy loss:

$$\mathcal{L}_{cls}^i = -\sum_{j=1}^{k} y_i \cdot \log(p_{ij}) \tag{20}$$

where $p$ is the predicted score of the classifier and $y$ is the sample label or generated pseudo-label. The classification loss and graph matching loss of this framework are trained in an end-to-end fashion to achieve more accurate classification result using holistic–local features and graph prototypes. The final optimization objective loss function is defined as follows:

$$\mathcal{L}^i = \mathcal{L}_{cls}^i + \lambda_{pro} \cdot \mathcal{L}_{pro}^i \tag{21}$$

Here, $\mathcal{L}_{cls}^i$ is the Categorical cross-entropy loss, $\mathcal{L}_{pro}^i$ is the graph matching loss, and $\lambda_{pro}$ is the Weight coefficient of the Categorical cross-entropy loss.

### 3.6. Implementation Details

#### 3.6.1. Network Architecture

Referring to [11], we also use the ResNet50-variant [12] as our backbone CNN, which consists of four block layers to extract features. Given an input image of size 448 × 448, we can obtain a feature map of 14 × 14 × 2048 from the fourth layer. For holistic features, we transform it into a feature vector of 2048 dimensions by adding an average pooling layer. For local features, we first use attention-focus to obtain nine regions around a discriminative area on a feature map, then adopt bilinear pooling to transform them into a fixed size 14 × 14 × 2048. Finally, an average pooling layer is used to transform them into feature vectors of 2048 dimensions. The fully connected layer with a single output channel of 512 is used as the edge feature embedding layer $f_{edge}()$. For the graph matching module, two fully connected layers are used as the graph propagation layer $f_{\theta_p}()$ and the output channels are 1024 and 2048, respectively. Then, a fully connected layer with a

single output channel number of 2048 is used as the graph update layer $f_{\theta u}()$. Finally a fully connected layer with a single output channel number of 2048 is used as the graph aggregation layer $f_{\theta_a}()$.

### 3.6.2. Training Details

The experiment is programmed using the PyTorch deep learning framework in python 3.6 under the Linux system. In terms of hardware, the CPU is the Intel Core i7-7800X and the GPU is the GTX 1080 Ti. We train the GRML framework in two stages. We initialize the parameters of the backbone with pre-trained parameters from the ImageNet dataset and the parameters of other layers are initialized randomly. In the first stage, we train the entire framework using the original labels. During training, we perform optimization using the stochastic gradient descent (SGD) algorithm with a batch size of 16, a momentum of 0.9 and a weight decay of 0.0001. We use the cosine annealing learning rate and the initial learning rate is set to 0.001. It is trained with 20 epochs. In the second stage, we will correct the noisy labels and remove the OOD samples, while the removed OOD samples will not participate in the final loss calculation. The temperature parameter $\tau$ is set to 0.1. The epoch is set to 80 in the second stage while other hyperparameters are the same as the first stage.

## 4. Experiments

### 4.1. Datasets

We conduct experiments on the benchmark dataset WebFG-496 [1] as the noisy source data which consist of three sub-datasets: Web-Bird, Web-Aircraft and Web-Car. The categories of these three webly supervised fine-grained benchmark sub-datasets correspond to CUB200-2011 [13], FGVC-Aircraft [14] and Stanford Cars [15]. The training sets of WebFG-496 were obtained from the web retrieval of the corresponding categories, while the validation sets in CUB200-2011, FGVC-Aircraft and Stanford Cars were used as the validation data. The details of the dataset can be seen in Table 1.

**Table 1.** This is the number of categories, the number of images and the estimation accuracy(%) of training images for each subdataset, where the estimation accuracy is estimated using a subset of random sampling.

| Dataset | Sub-Dataset | Classes | Training Image | Dataset Accuracy | Verifying Image |
|---|---|---|---|---|---|
| | Web-Bird | 200 | 18388 | 65 | 5794 |
| WebFG-496 | Web-Aircraft | 100 | 13503 | 73 | 3334 |
| | Web-Car | 196 | 21448 | 67 | 8041 |

### 4.2. Comparisons with the Existing Algorithms

The proposed GRML framework is compared with the existing representive models on three datasets. To conduct a comprehensive comparison, the comparison objects are divided into three categories. The first category is the basic models such as VGG-16 [12], ResNet-50 [16], GoogLeNet [17] and other models. This type of model does not process noisy data during training but treats them as a correct sample, which may mislead the training of the model. For example, the average accuracy of the best performing VGG-19 is only 68.63%. The second category is designed for web supervision scenarios represented by Decoupling [18], Co-teaching [19], Hanand PENCIL [20], which mainly deal with noisy data through network consensus among multiple networks. Compared with the basic model, this kind of model has an obvious improvement in classification performance. However, such models do not consider removing outlier samples that have the greatest impact on model training in noisy data. The third category is designed for web supervision fine-grained recognition scenarios represented by Peer-learning [1]. It mainly extracts overall features and processes noisy data through simple label consistency. Similarly, this model does not deal with outlier samples. Different from the above three types of models, the GRML framework proposed in this paper introduces graph prototypes and considers

the interaction of holistic–local information so as to more effectively correct noisy labels and remove outlier samples.

The comparison results are shown in Table 2. Compared with the existing methods, the GRML framework proposed in this paper has achieved superior performance on all datasets. The first is to compare with the basic model. The performance of the method in this paper on the three datasets is far better than that of various basic models. Considering that the backbone network of the framework proposed in this paper is ResNet-50. Compared with the separate ResNet-50 model, the method in this paper has greatly improved on the three datasets and the average accuracy rate has increased by 20.14%. For a fair comparison, we uniformly use ResNet-50 as the backbone network. From the experimental data, it can be seen that the method in this paper achieves a superior average accuracy of 81.77%, while the accuracy rates on Web-Bird, Web-Aircraft and Web-Car are 76.62%, 85.79% and 82.99%, respectively. Compared with the current more advanced method Peer-learning, it is 2.23%, 4.2% and 1.94% higher. Further, we use other models such as B-CNN as the backbone network, from the comparison results, it can be known that the GRML framework can be adapted to different backbone networks to obtain a relatively obvious performance improvement in web-supervised fine-grained recognition scenarios.

**Table 2.** Comparison of recognition accuracy(%) of different models on three datasets.

| Method | Backbone | Web-Bird | Web-Aircraft | Web-Car | Average |
|--------|----------|----------|--------------|---------|---------|
| ResNet-50 | - | 64.43 | 60.79 | 60.64 | 61.95 |
| ResNet-101 | - | 66.74 | 63.46 | 65.51 | 65.24 |
| VGG-16 | - | 66.34 | 68.38 | 61.62 | 65.45 |
| VGG-19 | - | 67.69 | 70.99 | 67.21 | 68.63 |
| GoogLeNet | - | 66.01 | 66.02 | 65.87 | 65.97 |
| B-CNN | - | 66.56 | 64.33 | 67.42 | 66.10 |
| Decoupling | B-CNN | 70.56 | 75.97 | 75.00 | 73.84 |
| Co-teaching | B-CNN | 73.85 | 72.76 | 73.10 | 73.24 |
| Peer-learning | B-CNN | 76.48 | 74.38 | 78.52 | 76.46 |
| Ours | B-CNN | 76.43 | 82.72 | 80.81 | 79.98 |
| Peer-learning | ResNet-50 | 74.15 | 81.07 | 81.40 | 78.87 |
| Ours | ResNet-50 | 76.62 | 85.79 | 82.99 | 81.77 |

### 4.3. Ablative Study

The core module of the GRML framework is the graph matching module. To analyze the impact of this module on the overall performance, two experiments are designed in three datasets, respectively. The first experiment (Ours w/o GPL) removes the graph propagation layer that mines the information interaction between nodes in the graph, which directly uses holistic and local features without propagation of intra-graph node information. The second group of experiments (Ours w/o GIL) removes the graph interaction layer for mining the similarity of information between graphs and it does not propagate information between graphs. The experimental results are shown in Table 3. After removing the graph propagation layer, the model will not explore the interaction between the holistic and local information and its performance on the three datasets is reduced by 2.17%, 3.38% and 2.75%, respectively. It is proven that making full use of the information interaction between holistic and local features can lead to significant performance improvement. At the same time, it can be noticed that even if the graph propagation layer is removed, its performance is still much better than using the ResNet-50 network alone, thus illustrating the importance of local features in fine-grained classification. After removing the graph interaction layer in the second experiment, the performance on the three datasets decreased by 1.4%, 1.65% and 1.05%, respectively. Since the purpose of the metric learning mechanism is to make the feature expressions of the same category close to each other, while different categories are far from each other, Therefore, when measuring the graph-level similarity, if

the information between graphs is not propagated, the result of the measurement will be unreliable, thus misleading the model training.

**Table 3.** Accuracies(%) of our GRML, our GRML that removes the graph interaction layer (Ours w/o intra-GPL) and our GRML that removes the graph interaction layer (Ours w/o GIL) on the three datasets.

| Method | Web-Bird | Web-Aircraft | Web-Car |
|---|---|---|---|
| Ours w/o GPL | 74.45 | 82.41 | 80.24 |
| Ours w/o GIL | 75.22 | 84.14 | 81.04 |
| Ours | 76.62 | 85.79 | 82.99 |

*4.4. Noisy Data Processing And Visualization*

In webly supervised learning, there is a lot of noise in the data set. The noise data can be divided into two categories, one is the noise label sample and the other is the outlier sample. These noise data will mislead the training of the model, so it is necessary to process these noisy data. To validate the effectiveness of the noise data processing method in this paper, we used the original web label for ablation experiments while the network structure remained unchanged. As shown in Table 4, the recognition accuracy on the three datasets was reduced by 3.56%, 5.90% and 7.33%, respectively. To further verify the effectiveness of noise processing, we randomly select some OOD samples and Non-OOD samples in each of the three datasets as shown in Figure 3. Compared with MoPro, the proposed GRML framework has similar performance in OOD sample rejection, while the GRML framework is superior in the prediction accuracy of Non-OOD samples. This illustrates how the proposed GRML framework can focus more on discriminative local features, so as to correct the noise labels and remove outlier samples according to the knowledge learned by the model, so as to alleviate the influence of noise data on model training.

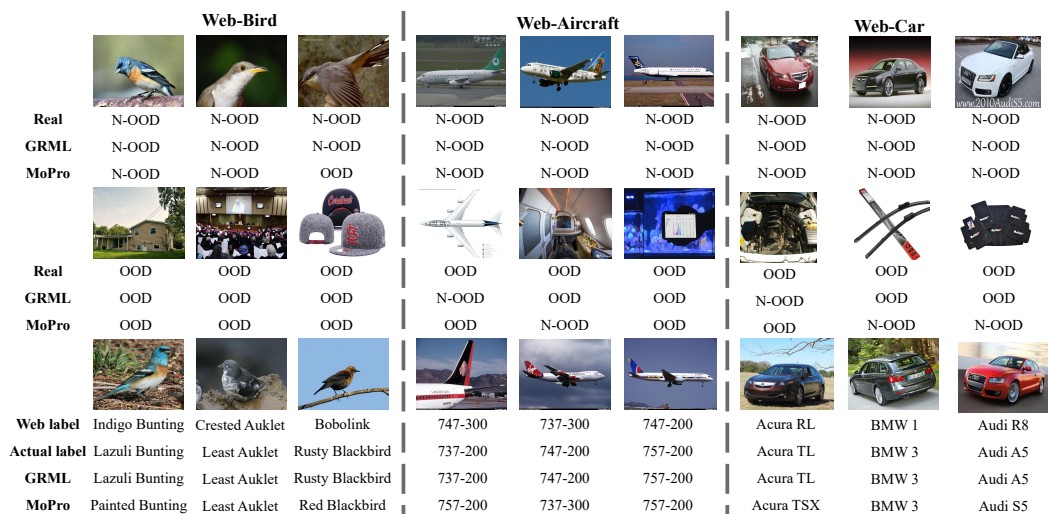

**Figure 3.** Example of prediction with GRML and MoPro for randomly selected Non-OOD, OOD and noisy label samples in each of the three datasets. Where "Real" represents whether the sample is actually OOD or not, "GRML" and "MoPro" indicate predicted or label corrected results.

**Table 4.** Noiselabel processing performance analysis.

| Method | Web-Bird | Web-Aircraft | Web-Car |
|---|---|---|---|
| Ours w/o correction | 73.65 | 82.41 | 78.23 |
| Ours | 76.62 | 85.79 | 82.99 |

*4.5. Conclusions*

In this work, we develop a novel graph representation and metric learning (GRML) framework that integrates graph propagation networks with prototype learning mechanisms that learn more discriminative holistic–local features to help correct the noisy labels and exclude OOD samples to facilitate webly supervised FIGR. Specifically, it consists of an attention-focused module that learns to locate local regions with the most discriminative content and a graph matching module to explore information interaction to learn instance-level and category-level graph representation, as well as their matching to correct the noisy labels and exclude OOD samples. We conduct extensive experiments on several online benchmarks to demonstrate the effectiveness of the proposed GRML framework.

**Author Contributions:** Conceptualization, J.L. (Jianman Lin), J.L. (Jiantao Lin) and Z.Y.; methodology, J.L. (Jianman Lin); software, J.L. (Jianman Lin); validation, Y.G., T.C. and J.L. (Jiantao Lin); formal analysis, J.L. (Jianman Lin), T.C.; investigation, J.L. (Jiantao Lin), Y.F; resources, J.L. (Jiantao Lin), T.C.; data curation, J.L. (Jiantao Lin); writing—original draft preparation, J.L. (Jianman Lin); writing—review and editing, Y.G., Z.Y.; visualization, J.L. (Jianman Lin); supervision, T.C., Y.G.; project administration, T.C., Y.G.; funding acquisition, Z.Y.; All authors have read and agreed to the published version of the manuscript.

**Funding:** This research was funded by the National Natural Science Foundation of China (NSFC), grant number 62206060, Natural Science Foundation of China Foundation of Fire, grants number HHJJ-2022-0102 and HHJJ-2022-0106, Science and Technology Project of Guangdong Province, grant number 2021A1515011341, and Guangdong Provincial Key Laboratory of Human Digital Twin, grant number 2022B1212010004.

**Data Availability Statement:** The data that support the findings of this study are openly available in https://github.com/NUST-Machine-Intelligence-Laboratory/weblyFG-dataset, accessed on 10 April 2021.

**Acknowledgments:** This work was supported in part by the National Natural Science Foundation of China (NSFC) under Grant 62206060, in part by the Natural Science Foundation of China Foundation of Fire under Grants HHJJ-2022-0102 and HHJJ-2022-0106, in part by the Science and Technology Project of Guangdong Province under Grant 2021A1515011341, and in part by the Guangdong Provincial Key Laboratory of Human Digital Twin under Grant 2022B1212010004.

**Conflicts of Interest:** The authors declare no conflict of interest.

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
