# Peer review of "Webly Supervised Fine-Grained Image Recognition with Graph Representation and Metric Learning"

_electronics, doi:10.3390/electronics11244127_

Round 1
Reviewer 1 Report
Authors present the develop a novel graph representation and metric learning (GRML) framework that integrates graph propagation networks with prototype learning mechanisms that learns more discriminative holistic-local features to help correct the noisy labels and exclude OOD samples to facilitate webly supervised FIGR. and evaluate its performance.
The paper is interesting at overall, as it considers a new training strategy for image recognition from web. Also, presented solution is complete and clear, with all the technical details regarding to the proposed method.
More information could be provided, such as the mainly data and how this method collect all those images. Also, specified the hardware were used for the process. Also, provide the pubic repository with the data.
The presented work is overall interesting, the technical details useful and challenging and the potential outcome of considerable impact.
In conclusion, the work is accepted, with the provision that the authors correct the point was mentioned previous.
Author Response
Response to Reviewer 1 Comments
Point 1: The mainly data and how this method collect all those images.
Response 1: Thank you for the comment. In this work, the main data source is the benchmark dataset WebFG-496 https://github.com/NUST-Machine-Intelligence-Laboratory/weblyFG-dataset, which consists of three sub-datasets containing a total of 53,339 web training images with 200 categories of birds (Web-bird), 100 categories of aircrafts (Web-aircraft), and 196 categories of cars (Web-car). We have added more detailed data information in Table 1 of the experiment part in Lines 295-296. In addition, the training set is obtained from web retrieval. It collects candidate training images from the web, and then removes broken images and duplicate images.
Point 2: Provide the hardware were used for the process.
Response 2: Thank you very much for your kind suggestion. In terms of hardware, the CPU is the Intel Core i7-7800X, and the GPU is the GTX 1080 Ti. We've added this information in the training details section in Lines 276–278.
Point 3: Provide the pubic repository with the data.
Response 3: Thank you very much for your kind suggestion. We have provided data information at the end of the paper on Line 388.
We sincerely thank your constructive feedbacks, which have significantly improved the quality of our manuscript.

Reviewer 2 Report
This work is about image recognition, a widely studied topic in ML using graph representation learning. I have few queries and comments below:
1.Page 10 table 2, the caption is missing 2. The methodology section is very difficult to follow. Please rewrite this portion by leaving out non-relevant text and focussing on important contributions. It is very difficult to follow the math. 3. The datasets used are limited 4. How do you handle long range information propagation in the graph network. Consider ceiling works such as this and explain how techniques such as long range information propagation can improve your work. This would improve the quality of the paper. M. K. Matlock, A. Datta, N. L. Dang, K. Jiang and S. J. Swamidass, "Deep learning long-range information in undirected graphs with wave networks," 2019 International Joint Conference on Neural Networks (IJCNN), 2019, pp. 1-8, doi: 10.1109/IJCNN.2019.8852455.Author Response
Response to Reviewer 2 Comments
Point 1: Page 10 table 2, the caption is missing.
Response 1: Thank you very much for your kind reminder. We have revised the mentioned issue.
Point 2: The methodology section is very difficult to follow. Please rewrite this portion by leaving out non-relevant text and focussing on important contributions. It is very difficult to follow the math.
Response 2: Thank you very much for your kind suggestion. We have revised the mentioned issue in attention module and graph matching module, and have thoroughly polished the representation and writing.
Point 3: The datasets used are limited.
Response 3: Thank you for the comment. The dataset is a widely available benchmark, which has a generally usable interface. Moreover, WebiNat-5089 is a very large dataset, which contains 5,089 sub-categories and more than 1.1 million web training images, and unfortunately, we don't have enough resources to run it.
Point 4: How do you handle long range information propagation in the graph network.
Response 4:Thank you for the suggestion. In this paper we only consider short-range information propagation. In our experiment, each graph structure has only 10 nodes, and any two nodes in the graph structure are connected to each other, so only short-range information propagation needs to be considered. Moreover, we have also modified the graph structure style in the overall framework; the previous graph structure style may be misleading, but in fact any two nodes are connected to each other.
We sincerely thank your constructive feedbacks, which have significantly improved the quality of our manuscript.

Reviewer 3 Report
The article is interesting and generally, it deserves to be published with some revisions that are suggested below:
1. In the Abstract, Would you please state what are the superior performances in short for the proposed approach when compared with current algorithms, and how to prove them?
2. In section 3( Experiments), Could you explain how many samples were run, tested, verify, and trained?
Author Response
Response to Reviewer 3 Comments
Point 1: In the Abstract, Would you please state what are the superior performances in short for the proposed approach when compared with current algorithms, and how to prove them?
Response 1: Thank you very much for your kind suggestion. To validate the proposed algorithm, we conducted extensive and fair experiments on three webly supervised FGIR benchmark datasets: Web-Bird, Web-Aircraft, and Web-Car. On Web-Bird, Web-Aircraft, and Web-Car, it achieves classification accuracy of 76.62%, 85.79%, and 82.99%, respectively. In comparison to Peer-learning, the classification accuracy of the three datasets improved by 2.47%, 4.72%, and 1.59%, respectively. In addition, we also used different backbones to conduct extensive comparative experiments on the above three datasets to demonstrate the effectiveness of the proposed framework, and the specific experimental effect is shown in Table 2 on page 10.
Point 2: In section 3( Experiments), Could you explain how many samples were run, tested, verify, and trained?
Response 2: Thank you for the comment. In Section 3 (Experiments), we added a table to explain how many samples were run, tested, verified, and trained (page 9).
We sincerely thank your constructive feedbacks, which have significantly improved the quality of our manuscript.

Round 2
Reviewer 2 Report
Authors addressed comments adequately